# Retrograde Axonal Transport of Liposomes from Peripheral Tissue to Spinal Cord and DRGs by Optimized Phospholipid and CTB Modification

**DOI:** 10.3390/ijms23126661

**Published:** 2022-06-15

**Authors:** Takafumi Fukui, Hironao Tateno, Takashi Nakamura, Yuma Yamada, Yusuke Sato, Norimasa Iwasaki, Hideyoshi Harashima, Ken Kadoya

**Affiliations:** 1Department of Orthopaedic Surgery, Faculty of Medicine and Graduate School of Medicine, Hokkaido University, Kita-15 Nishi-7, Sapporo 060-8638, Japan; takafumimed@gmail.com (T.F.); niwasaki@med.hokudai.ac.jp (N.I.); 2Faculty of Pharmaceutical Sciences, Hokkaido University, Kita-12 Nishi-6, Kita-ku, Sapporo 060-0812, Japan; tateno1670@eis.hokudai.ac.jp (H.T.); u-ma@pharm.hokudai.ac.jp (Y.Y.); y_sato@pharm.hokudai.ac.jp (Y.S.); harasima@pharm.hokudai.ac.jp (H.H.)

**Keywords:** liposome, drug-delivery system, retrograde axonal transport, motor neuron, DRG

## Abstract

Despite recent advancements in therapeutic options for disorders of the central nervous system (CNS), the lack of an efficient drug-delivery system (DDS) hampers their clinical application. We hypothesized that liposomes could be optimized for retrograde transport in axons as a DDS from peripheral tissues to the spinal cord and dorsal root ganglia (DRGs). Three types of liposomes consisting of DSPC, DSPC/POPC, or POPC in combination with cholesterol (Chol) and polyethylene glycol (PEG) lipid were administered to sciatic nerves or the tibialis anterior muscle of mature rats. Liposomes in cell bodies were detected with infrared fluorescence of DiD conjugated to liposomes. Three days later, all nerve-administered liposomes were retrogradely transported to the spinal cord and DRGs, whereas only muscle-administered liposomes consisting of DSPC reached the spinal cord and DRGs. Modification with Cholera toxin B subunit improved the transport efficiency of liposomes to the spinal cord and DRGs from 4.5% to 17.3% and from 3.9% to 14.3% via nerve administration, and from 2.6% to 4.8% and from 2.3% to 4.1% via muscle administration, respectively. Modification with octa-arginine (R8) improved the transport efficiency via nerve administration but abolished the transport capability via muscle administration. These findings provide the initial data for the development of a novel DDS targeting the spinal cord and DRGs via peripheral administration.

## 1. Introduction

Recent advancements in neuroscience have clarified the etiology of intractable CNS disorders and identified target molecules, leading to the development of therapeutic agents for such disorders [1,2,3,4,5,6]. However, due to the peculiarities of the CNS, the delivery of therapeutic agents to target tissues and cells in the CNS is difficult, hampering their clinical application [7]. For example, following oral or blood administration, most drugs poorly penetrate the blood–brain–spinal cord barrier, and very little drug reaches the parenchyma of the CNS [8,9]. Administration in cerebrospinal fluid has low tissue specificity and involves risks associated with the procedure, including infection and hematoma [10,11]. Local injection has the advantage of specific delivery of drugs to a target region but requires surgical procedures, which involve invasiveness and limit the delivery site and frequency [12]. Thus, the development of a minimally invasive, cell-specific, frequently applicable DDS for the CNS is needed.

In the last few decades, efforts have been made toward the development of viral vectors for retrograde transport from peripheral tissue to the spinal cord via axons [13,14,15]. Although new serotypes of adeno-associated virus (AAV) improved their retrograde axonal transport efficiency, many investigations used postnatal subjects [16,17,18,19], in which transport efficiency is higher than in adult subjects. As far as we know, the maximum transport efficiency of AAV from muscle to spinal cord is 5% of spinal cord motor neurons per total projecting neurons [13,20,21], and further improvement is required. In addition, viruses have a limitation in which they can only be used to deliver nucleic acids. Accordingly, the development of an efficient DDS to deliver not only nucleic acids but also proteins and chemicals in a retrograde axonal transport manner is desired [13,22].

To develop a novel DDS for retrograde axonal transport, we employed liposomes that consist of hydrophilic and hydrophobic molecules and that can efficiently encapsulate nucleic acids, proteins, and chemical compounds for delivery into cells [23,24]. Liposomes have already been clinically used as a DDS in various fields, such as therapeutics for malignant tumors and infectious diseases and vaccines for viral infections, indicating their great biocompatibility and safety [25,26,27]. An additional advantage is that they can be optimized for cellular uptake efficiency and intracellular behavior by controlling their lipid composition and surface modification [28,29,30,31]. A previous attempt on axonal retrograde transport of liposomes was an administration of Cholera toxin B subunit (CTB)-conjugated protocells into pleura, demonstrating their retrograde axonal transport in axons but not the spinal cord [32]. To the best of our knowledge, no study has showed retrograde axonal transport of liposomes from peripheral tissues to neuronal cell bodies.

CTB and Fluoro-Gold (FG) are well-known to undergo efficient retrograde transport in axons and have a long history as neuronal tracers in neuroscience research [33,34,35,36,37]. A few days are needed for CTB and FG to be transported from the sciatic nerve or hindlimb muscles to motor neurons in the spinal cord in rodents [38,39,40], indicating that liposomes need to have sufficient biostability in axons until they reach the cell body (Figure 1a). Thus, we hypothesized that optimized liposomes could be taken up into axons and efficiently transported from peripheral tissues to the CNS in a retrograde manner. The purpose of the current study is to develop liposomes that are retrogradely transported from peripheral nerves or muscles to the spinal cord or dorsal root ganglia (DRGs) by optimizing the lipid composition and modifying the surface (Figure 1a,b).

## 2. Results

### 2.1. Liposomes Consisting of DSPC, Chol, and PEG Lipid Are Retrogradely Transported from Peripheral Nerves to the Spinal Cord and DRG

Since the cellular uptake efficiency and intracellular behavior of liposomes depend on the composition of phospholipids [41,42,43,44], we evaluated different phospholipid compositions for retrograde axonal transport. Considering that sufficient biostability of liposomes is needed for several days in axons, we prepared three types of liposomes consisting of Chol, N-(Carbonyl-methoxypolyethyleneglycol 2000)-1,2-distearoyl-sn-glycero-3-phosphoethanolamine (DSPE-PEG2000), 1,2-Distearoyl-sn-glycero-3-phosphocholine (DSPC), and 1-Palmitoyl-2-oleoyl-sn-glycero-3-phosphocholine (POPC), based on a previous study showing that DSPC-based liposomes (DSPC-Lip) have high biostability [45]. The liposomal membrane composed of DSPC shows high stability, because DSPC has the saturated chain and a longer alkyl chain compared with POPC. DSPC-Lip, POPC-Lip, and DSPC/POPC-Lip were prepared and labeled with 1,1′-dioctadecyl-3,3,3′,3′-tetramethylindodicarbocyanine, 4-chlorobenzenesulfonate (DiD). The characteristics of these liposomes are described in Table 1.

Adult rats received injections of 2 μL of liposomes into sciatic nerves (Figure 2a), followed by perfusion 3 days later. When examining spinal cord sections, the fluorescent signal of DiD was detected in spinal motor neurons in all liposome groups (Figure 2b). The total number of DiD-labeled motor neurons was divided by the total number of CTB-labeled motor neurons to calculate the proportion of DiD-labeled neurons. DSPC-Lip demonstrated the highest efficiency, which was 4.5%, followed by DSPC/POPC-Lip with 3.2%, and POPC-Lip with 0.96% (Figure 2c). Importantly, injections of DiD alone did not reach the spinal cord (Figure 2b,c), indicating that DiD cannot be transported to the spinal cord like CTB and that the fluorescent signal of DiD in motor neurons is not a consequence of detachment of DiD from the liposomes.

Since the sciatic nerve consists of axons from spinal motor neurons and DRG neurons, liposomes injected into the sciatic nerve may be transported to DRGs as well. Thus, we next examined the presence of DiD-labeled neurons in L5 DRGs. Similar to the observation in the spinal cord, the fluorescent DiD signal was detected in DRGs in rats administered each of the three types of liposomes (Figure 2d,e). Quantification demonstrated that the proportion of DiD-labeled DRG neurons was the highest in rats receiving DSPC-Lip at 3.9%, followed by DSPC/POPC-Lip with 2.6%, and POPC-Lip with 0.8% (Figure 2e), similar to the results in the spinal cord (Figure 2c). These findings show that liposomes consisting of DSPC, POPC, Chol, and PEG lipid can be transported from peripheral nerves to the spinal cord and DRGs by retrograde axonal transport, that the transport efficiency depends on the proportion of DSPC, and that transport efficiency does not differ between the spinal cord and DRGs.

### 2.2. Liposomes Consisting of DSPC, Chol, and PEG Lipid Are Retrogradely Transported from Muscle to Spinal Cord and DRGs

Next, we determined whether the liposomes developed as described above could be retrogradely transported from muscle to the spinal cord and DRGs. Adult rats received injections of 2 μL of liposomes into the tibialis anterior muscles (Figure 3a), followed by perfusion 3 days later. Evaluation of spinal cord sections demonstrated that rats receiving DSPC-Lip showed DiD fluorescence in spinal motor neurons, whereas rats receiving other liposomes did not. The proportion of DiD-labeled motor neurons among the total CTB-labeled spinal motor neurons was 2.6% in rats administered DSPC-Lip (Figure 3b,c). As with the spinal cord findings, evaluation of DRGs demonstrated that rats receiving DSPC-Lip showed DiD fluorescence in DRGs, whereas other rats did not. The proportion of DiD-labeled DRG neurons per total FG-labeled DRG neurons was 2.3% in rats administered DSPC-Lip (Figure 3d,e). These findings indicate that liposomes consisting of DSPC but not POPC with Chol and PEG lipid can be retrogradely transported from muscle to the spinal cord and DRGs, and that no difference in transport efficiency was apparent between spinal motor neurons and DRG neurons following muscle administration.

### 2.3. Modification with R8 and CTB Improves Uptake of Liposomes by Cultured Spinal Motor Neurons

Modification of liposomes affects their behaviors as a DDS, such as uptake efficiency, intracellular stability, and endosomal escape [30,46,47,48,49]. CTB is a pentameric protein that forms a part of Cholera toxin, is not cytotoxic, enters cells via the monosialotetrahexosylganglioside (GM1) receptor, and is transported retrogradely in axons to neuronal cell bodies [29,50]. In addition, previous studies revealed that modification of liposomes with R8 greatly promotes their cellular uptake [49,51,52,53]. Accordingly, we hypothesized that the efficiency of retrograde axonal transport from peripheral tissue to the spinal cord and DRGs could be further improved by modification with CTB and R8. We prepared four types of liposomes with CTB, R8, both, or neither, and all liposomes were conjugated with DiD as a fluorescent marker. The characteristics of these liposomes are described in Table 1. The efficiency of cellular uptake of each type of liposome was examined using a cultured spinal cord motor neuron-like cell line, NSC34 [29,54]. Prepared liposomes were added to the culture medium, followed by quantification of the cells labeled with DiD 2 h later. Control liposomes with no modification (DSPC-Lip) did not enter cells at the tested concentrations (Figure 4a,b), whereas the liposomes modified with CTB or R8 entered the cells. In particular, R8-modified liposomes demonstrated the most efficient cellular uptake among the tested groups. At the concentration of 0.14 mM of phospholipids, R8-modified liposomes demonstrated 100% cellular uptake. CTB-R8-modified liposomes showed 100% cellular uptake at 0.42 mM of phospholipids, and were superior to CTB-modified liposomes (Figure 4a,b). These results indicate that modification with R8 and CTB is effective for uptake of DSPC liposomes by spinal motor neurons and that R8 is particularly effective. Of note, modification with CTB impaired the effect of R8, suggesting the presence of an inhibitory interaction between R8 and CTB. Importantly, although control liposomes without modification were retrogradely transported in axons following sciatic nerve administration (Figure 2b–e) and tibialis anterior muscle administration (Figure 3b–e), these cultured motor neurons did not take up the liposomes, indicating that this in vitro assay does not necessarily mimic retrograde axonal transport in vivo.

### 2.4. Modification with CTB Improves Retrograde Axonal Transport of Liposomes from Peripheral Nerves to the Spinal Cord and DRGs

Next, we determined whether R8 and CTB modification improved the retrograde axonal transport of liposomes via nerve administration. We expected that modification of CTB or R8 would improve the transport efficiency, and that modification of both CTB and R8 would have synergistic or additive effects. Similar to the preceding experiment, 2 μL of four types of liposomes with or without R8 and CTB modification were administered to rat sciatic nerves, followed by perfusion three days later. All types of liposomes reached and accumulated at spinal motor neurons (Figure 5a,b), further confirmed by the presence of immunoreactivity against ChAT (Figure 5c). The proportion of DiD-labeled spinal motor neurons was the highest for CTB-modified liposomes at 17.3%, followed by CTB-R8-modified liposomes at 13.0%, R8-modified liposomes at 9.6%, and control liposomes at 4.4% (Figure 5a,d), showing that CTB, CTB-R8, and R8 modification improved the transport efficiency by about 3.9, 3, and 2.2 times compared to the control liposomes, respectively. When evaluating transport 10 days after administration, CTB modification still demonstrated the highest proportion of DiD-labeled spinal motor neurons, 26.7%, followed by CTB-R8, R8, and the control, showing the same order as the result after 3 days (Figure 5e). The transport efficiency increased by about 20% to 60%, compared to that at 3 days.

Next, retrograde transport of these liposomes to DRGs was evaluated. The proportion of DiD-labeled DRG neurons 3 days after administration was the highest, 14.3%, for CTB-modified liposomes, 10.6% for CTB-R8-modified liposomes, 7.3% for R8-modified liposomes, and 3.7% for the control (Figure 5f,g), showing the same order as that of spinal motor neurons. Examination 10 days after administration showed an increased proportion of DiD-labeled DRG neurons. CTB modification produced the highest proportion at 23.4%, followed by CTB-R8, R8, and the control (Figure 5h). These results indicate that modification of liposomes with CTB and R8 improves their efficiency of retrograde axonal transport from peripheral nerves to the spinal cord and DRGs, that the modification effect was not specific to neuronal types, and that the transport efficiency was improved in a time-dependent manner, at least until 10 days after administration. Notably, CTB modification was quite effective; however, no synergistic effect of R8 and CTB was detected. Rather, R8 modification inhibited the effect of the CTB modification, in contrast to the result with motor neuron cultures in the preceding experiment.

### 2.5. Modification with CTB Improves Retrograde Axonal Transport of Liposomes from Muscles to the Spinal Cord and DRGs

Next, we determined whether R8 and CTB modification also improves the retrograde axonal transport of liposomes in sciatic nerves via muscle administration. Similar to the preceding experiment, adult rats received injections of 2 μL of liposomes into tibialis anterior muscles, followed by perfusion 3 days later. The fluorescent DiD signal was detected in spinal motor neurons in rats receiving control liposomes and CTB-modified liposomes, whereas rats receiving R8- or CTB-R8-modified liposomes did not show any fluorescent DiD signal in spinal cords (Figure 6a). The proportion of DiD-labeled spinal cord motor neurons was 4.8% with CTB-modified liposomes, which was about 1.8 times that of control liposomes (Figure 6b). Ten days after administration, rats receiving R8- and CTB-R8-modified liposomes still failed to show the fluorescent DiD signal in spinal cords, whereas rats receiving control liposomes and CTB-modified liposomes demonstrated an increased proportion of DiD-labeled spinal motor neurons, a 2.5- and 2.9-fold increase, respectively, compared to that three days after administration (Figure 6c).

When DRGs 3 and 10 days after administration were evaluated, similar to the results with the spinal cord, rats receiving control liposomes and CTB-modified liposomes showed the fluorescent DiD signal in neurons, whereas rats receiving R8- and R8-CTB-modified liposomes failed to show fluorescent DiD signals (Figure 6d–f). The percentages of DiD-labeled neurons 3 and 10 days after administration were 4.1% and 7.3% for CTB-modified liposomes and 2.3% and 3.2% for control liposomes, respectively (Figure 6e,f). These results indicate that CTB modification significantly improves retrograde axonal transport of liposomes from muscles to the spinal cord and DRGs. In addition, R8 modification, which is effective following nerve administration, impairs the retrograde axonal transport of liposomes from muscle, suggesting that the mechanism of liposome uptake greatly differs between nerve and muscle.

## 3. Discussion

The current study revealed that liposomes consisting of phospholipid, Chol, and PEG lipid can be retrogradely transported from peripheral tissues to the spinal cord and DRGs. Furthermore, the efficiency of axonal transport of liposomes can be enhanced by optimizing the lipid composition and modification with CTB.

Regarding retrograde axonal transport, axons utilize two types of transport. One is fast transport via dynein, and the other is slow movement via diffusion. In rats, the former is 73 mm/day or faster [55], and the latter is 0.2–0.6 mm/day [56,57], meaning that about 1.4 days are required for the former, and 17 to 50 days are needed for the latter for transport from the lower limb muscles to the spinal cord in adult rats. Therefore, liposomes need to be loaded onto the fast axonal transport system when applied as DDS. To induce retrograde axonal transport of liposomes, three processes are needed: liposomes must be taken up into axons, loaded on dynein, and transported to the neuronal cell body via dynein. This is in sharp contrast to most liposome applications as a DDS in which liposomes are directly taken up at the cell body. In the current study, the efficiency of retrograde transport from peripheral tissues to neuronal cell bodies greatly differed depending on the method of administration, lipid composition, and surface modification, indicating that these properties significantly affect uptake into axons, loading onto dynein, and stability in axons and on dynein.

Regarding the mechanism of uptake of liposomes into axons, the molecular functions of axons differ from those of neuronal cell bodies to some extent [58,59]. At the same time, the surface of axons is composed of a lipid bilayer similar to that of cell bodies [58]. Further, axons share the same endocytosis mechanisms as cell bodies, including macropinocytosis and clathrin-dependent endocytosis [60]. Thus, the mechanism of liposome uptake into axons appears to be nearly the same as that for cell bodies. On the other hand, in the case of intramuscular administration, liposomes have to be taken up at axon terminals at the neuromuscular junction, sensory receptors, and free axonal endings [61,62]. Since axon terminals have specific anatomical features [63], such as synaptic clefts where vesicles containing neurotransmitters are released and taken up [64,65], the mechanism of liposome uptake at nerve endings seems to differ from that at cell bodies.

Dynein actively and retrogradely transports intracellular cargo such as vesicles and endosomes, which have a similar structure to liposomes, to neuronal cell bodies [66,67]. To develop DDS for retrograde axonal transport, the loading efficiency on dynein needs to be improved, but the details of dynein loading remain unclear. Progress in understanding the mechanism of axonal cargo by dynein is required to develop an efficient DDS for retrograde axonal transport.

The current study showed that the efficiency of retrograde axonal transport of liposomes increased as the proportion of DSPC increased, perhaps because DSPC has lower membrane fluidity and greater biostability than POPC [68]. In addition, the lipid composition may affect the uptake of liposomes at axons and nerve endings [44]. Further analysis testing other lipid compositions such as 1,2-Dioctadecanoyl-sn-glycero-3-phosphoethanolamine (DSPE) and Dipalmitoylphosphatidylcholine (DPPC) via muscle administration may contribute to optimization of liposomes for retrograde axonal transport.

In the current study, CTB modification significantly increased the efficiency of retrograde axonal transport. Several mechanisms may underlie this improvement. CTB efficiently enters axons via GM1 receptors [69,70], and CTB-modified liposomes enter cultured cells via the same mechanism as CTB alone [29,30]. Thus, the uptake of liposomes at axons seems to be improved by CTB modification. In addition, CTB modification promotes endosomal escape [30,32], potentially contributing to reduced degradation in axons. Furthermore, CTB modification may increase the efficiency of liposome loading onto dynein [29].

R8, which is composed of eight arginines, has excellent cell permeability and increases cellular uptake of proteins and nucleic acids when conjugated to them [31,49]. For liposomes, R8 modification improves uptake by fibroblasts, HeLa cells, and MDCK cells [46,49,71,72]. This is partially because R8 modification increases the ζ potential of liposomes, enhancing the interaction with the cell surface, which is negatively charged with proteoglycans [73,74], or triggers macropinocytosis [49]. The current study demonstrated that R8 modification improved the retrograde axonal transport of liposomes when injected into nerves. Since R8 modification significantly enhanced uptake by cultured motor neurons (Figure 4a,b), R8 modification likely increases axonal uptake rather than improving the loading capability onto dynein and the biostability. In contrast, when injected into muscle, R8 modification completely abolished the capability of retrograde axonal transport of liposomes. Since R8 modification will also improve liposome uptake by muscle cells, most liposomes may enter muscle cells before being taken up at axon terminals. In particular, because axon terminals have specific anatomical features, as mentioned above [63], R8 modification may be ineffective for the improvement of the uptake efficiency at axon terminals. These findings suggest that retrograde axonal transport of liposomes from muscle may be impaired simply due to enhanced nonspecific cellular uptake. In addition, regardless of the administration route, the combined use of CTB and R8 modification failed to show synergistic or additive effects on the transport efficiency. Although it is unclear which process in axonal uptake, dynein loading or biostability, limits their combinatorial effect, a simple combination of modifications did not improve the transport efficiency of retrograde axonal transport of liposomes.

Recently, genes and molecules related to pathological conditions involving spinal cord motor neurons and DRG neurons have been elucidated. For example, downregulation of KCC2 in spinal cord motor neurons is involved in spasticity associated with spinal cord injury [75], and CLP257 and 290, which increase the expression of KCC2, are candidate therapeutic agents [6,76]. Further, *SOD1*, *C9orf72*, and *SMN1* have been identified as causative genes for amyotrophic lateral sclerosis and spinal muscular atrophy, in which the viability of spinal motor neurons is decreased [5,77], and treatment strategies targeting these genes are under development [78,79,80]. In addition, upregulation of NKCC1 in DRG neurons is involved in chronic pain, and bumetanide and furosemide suppress this upregulation, contributing to pain relief [3]. Although further improvement is required, retrogradely transported liposomes in axons have great potential as a minimally invasive, efficient, and target-specific DDS for these disorders. Importantly, the current study provides evidence that liposomes can be transported retrogradely in axons from peripheral tissues to the CNS, as observed by detection of DiD conjugated to the liposomes. Further research is necessary to demonstrate that an encapsulated substance in liposomes functions after transport to neuronal cell bodies in the spinal cord and DRGs.

## 4. Materials and Methods

### 4.1. Animals

Adult male and female LEWIS rats (10–14 weeks old, wild-type, Charles River Laboratories Japan, Inc.) were used in all experiments. Their body weight ranged from 160 to 220 g. Animals were group-housed with their littermates in a dedicated housing room under a 12 h light/12 h dark cycle and had free access to food and water throughout the study. For animal anesthesia, a mixture of ketamine (75–100 mg/kg, KETALAR^®^, Daiichi Sankyo Propharma Corporation, Tokyo, Japan) and medetomidine (0.5 mg/kg, DOMITOR^®^, Orion Corporation, Espoo, Finland) was administered by intraperitoneal injection.

### 4.2. Preparation of Liposomes

Lipids constituting liposomes were POPC (NOF Corporation, Tokyo, Japan), DSPC (NOF Corporation), Chol (Avanti Polar Lipids Inc., Alabaster, AL, USA), and DSPE-PEG2000 (NOF Corporation, Tokyo, Japan). Three types of lipid composition were employed: POPC/Chol/DSPE-PEG2000 = 70/30/5 (mol ratio), DSPC/POPC/Chol/DSPE-PEG2000 = 30/40/30/5 (mol ratio), and DSPC/Chol/DSPE-PEG2000 = 70/30/5 (mol ratio). DiD (Thermo Fisher Scientifc, Waltham, MA, USA) was added to lipid solutions for labeling liposomes fluorescently. Since DiD is lipophilic, it was incorporated into the hydrophobic layer of the liposomes. The lipids and DiD (0.5 mol% of total lipids) solved in chloroform were added to a glass tube, and the organic solvent was removed by N_2_ gas for a lipid film preparation. The lipid film was hydrated with 10 mM of HEPES buffer containing 5% glucose at a 5 mM lipid concentration. The liposome preparation was carried out with a bath-type sonicator. For R8 modification on the liposomes, stearylated R8 (STR-R8) (KURABO, Osaka, Japan) was added to the lipid chloroform solution. For CTB modification on the liposomal surface, 0.1 mol% of biotinylated DSPE-PEG2000 (NOF Corporation, Tokyo, Japan) was added to the lipid chloroform solution. After the liposome preparation, avidin and biotinylated CTB (Sigma-Aldrich, St. Louis, MO, USA) were added to the liposome preparation (biotinylated DSPE-PEG2000/avidin/biotinylated CTB = 1/1/1, mol ratio). A Zetasizer Nano (Malvern Institute Ltd., Malvern, WR, UK) was employed for evaluating the liposomal characters.

### 4.3. Surgical Procedures

For nerve injection, anesthetized rats were placed in a prone position, and the sciatic nerve was exposed by making a longitudinal skin incision (4 cm long) from buttock to distal thigh. A total of 2.0 µL of liposomes, described in Table 1, 0.023 mM of DiD, and 2.0 µL of 1% CTB (Sigma-Aldrich, St. Louis, MO, USA), 10% FG (Fluorochrome, Denver, CO, USA), were injected into the sciatic nerve using a Hamilton syringe (30-gauge needle, HAMILTON, Reno, NV, USA). For muscle injection, a 0.3 cm-long skin incision was made on the tibialis anterior muscle, followed by an injection of 2.0 µL of liposome solution, 0.023 mM of DiD, or 1% of CTB, 10% of FG into the middle of the muscle using a micro-syringe (30-gauge needle, HAMILTON, Reno, NV, USA). DiD was dissolved in ethanol first, and then adjusted to 0.023 mM in saline. CTB and FG were directly dissolved in saline.

### 4.4. Histology

The rats were transcardially perfused with 0.1 M phosphate buffer (PB, pH 7.4), followed by 4% paraformaldehyde (PFA, Nacalai Tesque Inc., Kyoto, Japan) in 0.1 M PB. The lumbar spinal cords and DRGs were dissected and postfixed in 4% PFA overnight at 4 °C, followed by transferring them into 30% sucrose in 0.1 M PB for cryoprotection. Spinal cords were transversely sectioned at 35 µm intervals using a microtome (REM-710; Yamato Kohki Industrial Co., Saitma, Japan). DRGs were sectioned at 14 µm intervals using a cryostat (Leica Biosystems CM3050, Wetzlar, Germany). In rats, the sciatic nerve is composed of the fourth, fifth, and sixth lumbar spine (L4, 5, and 6) nerve fibers, and the tibialis anterior muscle is mainly innervated by L4 nerve fibers [81,82], so we evaluated L5 DRG in the nerve administration group and L4 DRG in the muscle administration group.

### 4.5. Immunolabeling

For immunolabeling, sections were blocked with 10% horse serum (Thermo Fisher Scientific, Waltham, MA, USA) in 0.1 M TBS, followed by the overnight incubation with primary antibody against choline acetyltransferase (ChAT, goat, 1:200, AB144P; Sigma-Aldrich, St. Louis, MO, USA) or CTB (goat, 1:500, 703; List Biological Laboratories, Inc., Campbell, NJ, USA) at 4 °C. The next day, sections were incubated with donkey anti-goat conjugated to Alexa 594 (1:500, Jackson Immuno Research Laboratories, Inc, West Grove, PA, USA) for 2.5 h at room temperature.

### 4.6. Quantification

Quantification of liposome transport to the spinal cord and DRGs was performed as previously described [20]. In brief, one-of-six serial sections of lumbar spinal cord or one-of-ten sections of DRGs were imaged by an all-in-one fluorescent microscope (BZ-X710; Keyence, Osaka, Japan) with the same exposure time and ISO in the same comparison. DiD-, CTB-, or FG-labeled neurons were identified by the presence of the signals of 668, 518, or 460 nm wavelengths in neuronal soma but not the nucleus. The total number of fluorescently labeled neurons of all sections was counted. The transport efficiency was calculated by the total number of DiD-labeled neurons divided by the total number of CTB- or FG-labeled neurons.

### 4.7. Cell Culture

NSC-34 motor neuron-like cells [29] were purchased from CELLutions Biosystems Inc. and maintained in Dulbecco’s modified Eagle’s medium (DMEM, Wako, Osaka, Japan), supplemented with 10% fetal bovine serum (FBS) and 1% penicillin/streptomycin (PS, Thermo Fisher Scientific, Waltham, MA, USA). Cells were seeded at a density of 2.0 × 10^4^ cells/cm^2^ in 48-well plates and incubated with liposomes at 37 °C. During the liposome incubation, cell culture medium did not include FBS. Two hours later, cells were fixed with 4% PFA for 15 min, followed by 5 min incubation with 4’,6-diamidino-2-phenylindole (DAPI, 1:500, Sigma-Aldrich, St. Louis, MO, USA). Images were taken by an all-in-one fluorescent microscope (BZ-X710). At least 100 cells were randomly selected, and the number of DiD-labeled cells was counted to calculate the proportion of DiD-labeled cells per total number of cells.

### 4.8. Statistical Analysis

For statistical analyses, multiple-group comparisons were performed with one-way ANOVA and the Tukey–Kramer test. All analyses were performed with JMP software (SAS, Cary, NC, USA) with a pre-specified significance level of 95%. Data are presented as the mean ± standard deviation (SD).

## 5. Conclusions

As far as we know, the current study demonstrated for the first time that liposomes can be retrogradely transported from peripheral tissues to the spinal cord and DRGs in axons, that their transport efficiency depends on the lipid composition to enhance biostability, and that transport efficiency can be improved by modification with CTB for muscle and nerve administration and R8 for nerve administration. These results provide the basis for the development of a novel DDS targeting the spinal cord and DRGs via peripheral administration. Further efforts are required to improve the transport efficiency as well as the functionality as a DDS.

## Figures and Tables

**Figure 1 ijms-23-06661-f001:**
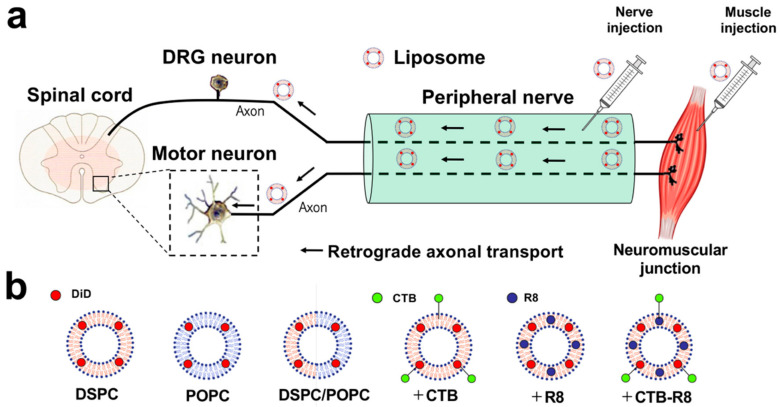
Schematic illustration of retrograde transport of liposomes from peripheral nerves or muscle to spinal cord motor neurons and DRG neurons. (**a**) Schematic overview of retrograde axonal transport of peripherally administered liposomes. (**b**) Schematic overview of liposomes consisting of different phospholipids and modified with CTB, R8, or both.

**Figure 2 ijms-23-06661-f002:**
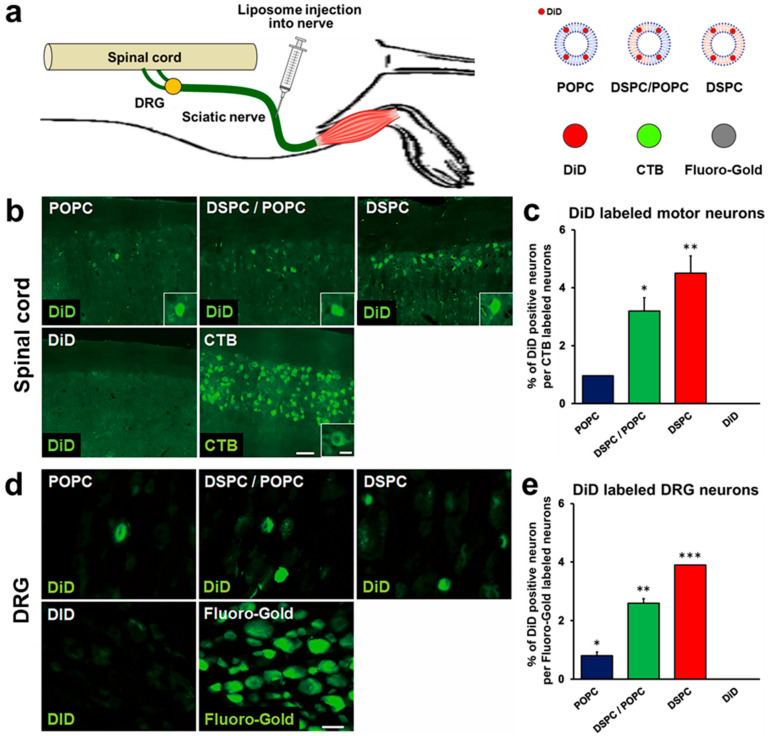
Retrograde transport of liposomes consisting of different phospholipids to spinal cord motor neurons and DRG neurons 3 days after sciatic nerve injection. (**a**) Schematic overview of sciatic nerve injection. (**b**) Horizontal sections of the lumbar spinal cord. Rostral is left. Liposomes were composed of POPC alone, DSPC alone, or both POPC and DSPC. Motor neurons receiving retrogradely transported liposomes were identified by fluorescence of conjugated DiD. Insets are high-magnification views of DiD-labeled motor neurons. DiD alone was not transported to motor neurons in 3 days, whereas many motor neurons were labeled by CTB. Scale bars, 100 µm, 30 µm (inset). (**c**) The efficiency of retrograde transport of liposomes. The percentage of DiD-labeled motor neurons of the total number of CTB-labeled motor neurons (*n* = 3/group). DSPC liposomes had the highest transport efficiency among the four groups. Data are the mean ± standard deviation (SD). * *p* < 0.05, ** *p* < 0.05 vs. all other groups by one-way ANOVA with the Tukey–Kramer test. (**d**) L5 DRG sections. DRG neurons receiving retrogradely transported liposomes were identified by DiD fluorescence. DiD alone was not transported to DRG neurons, whereas most neurons were labeled with Fluoro-Gold. Scale bar, 40 µm. (**e**) The percentage of DiD-labeledDRG neurons of the total number of Fluoro-Gold-labeled neurons was quantified (*n* = 3/group). Data are the mean ± SD. * *p* < 0.05, ** *p* < 0.05, *** *p* < 0.05 vs. all other groups by one-way ANOVA with the Tukey–Kramer test.

**Figure 3 ijms-23-06661-f003:**
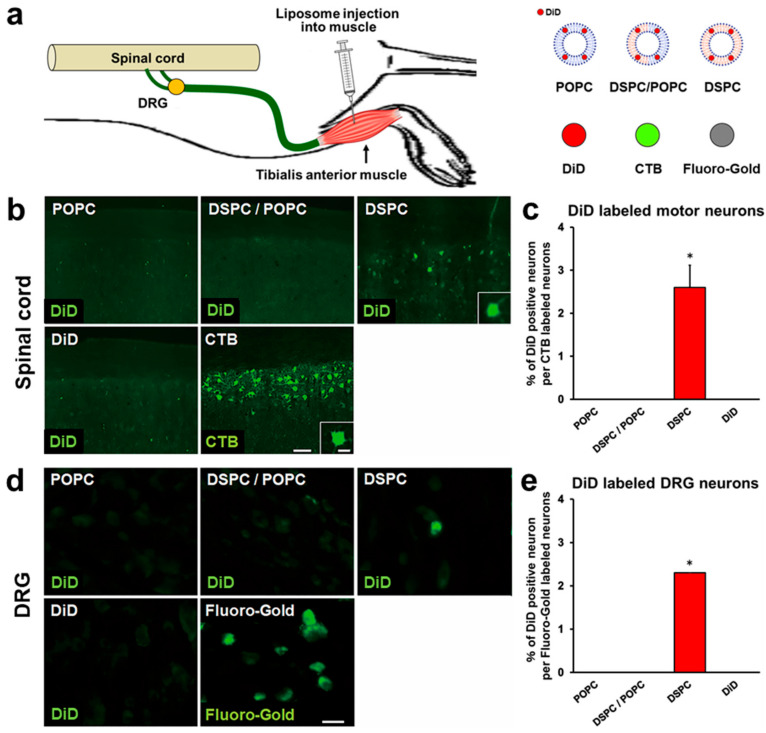
Retrograde transport of liposomes consisting of different phospholipids to spinal cord motor neurons and DRG neurons 3 days after tibialis anterior injection. (**a**) Schematic overview of tibialis anterior injection. (**b**) Horizontal sections of the lumbar spinal cord. Rostral is left. Motor neurons receiving retrogradely transported liposomes were identified by DiD fluorescence. Insets are high-magnification views of DiD-labeled motor neurons. Only DSPC liposomes were identified in motor neurons among the three types of liposomes. Scale bars, 100 µm, 30 µm (inset). (**c**) The efficiency of retrograde transport of liposomes. The percentage of DiD-labeled motor neurons of the total number of CTB-labeled motor neurons (*n* = 3/group). Data are the mean ± SD. * *p* < 0.05 vs. all other groups by one-way ANOVA with the Tukey–Kramer test. (**d**) L4 DRG sections. DRG neurons receiving retrogradely transported liposomes were identified by DiD fluorescence. Only DSPC liposomes were identified in DRG neurons among the three types of liposomes. Scale bar, 40 µm. (**e**) The percentage of DiD-labeledDRG neurons of the total number of Fluoro-Gold-labeled neurons was quantified (*n* = 3/group). Data are the mean ± SD. * *p* < 0.05 vs. all other groups by one-way ANOVA with the Tukey–Kramer test.

**Figure 4 ijms-23-06661-f004:**
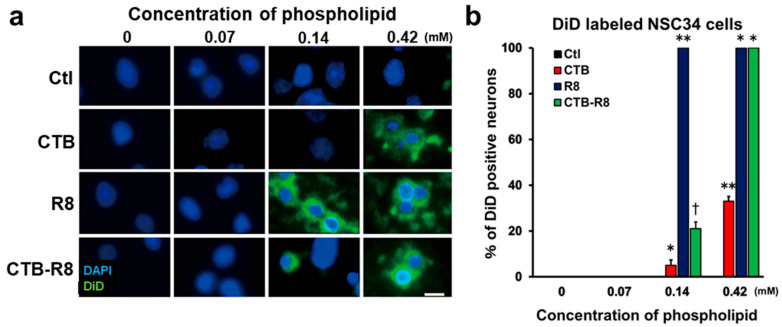
Uptake of liposomes modified with CTB and R8 by NSC34 motor neuron-like cells. (**a**) High-magnification images of NSC34 motor neuron-like cells cultured with CTB- and R8-modified DSPC liposomes. Cells taking up liposomes were identified by DiD fluorescence. Unmodified liposomes did not enter any cells, even at the high concentration, whereas R8- or CTB-modified liposomes were taken up by cells. Scale bar, 20 µm. (**b**) The percentage of DiD-labeled NSC34 motor neuron-like cells of the total number of cells. At least 100 cells per sample were quantified (*n* = 3 samples/condition). R8-modified liposomes were taken up by all cells at a concentration of 0.14 mM of phospholipids. Data are the mean ± SD. * *p* < 0.05, ** *p* < 0.05, † < 0.05 vs. all other groups by one-way ANOVA with the Tukey–Kramer test.

**Figure 5 ijms-23-06661-f005:**
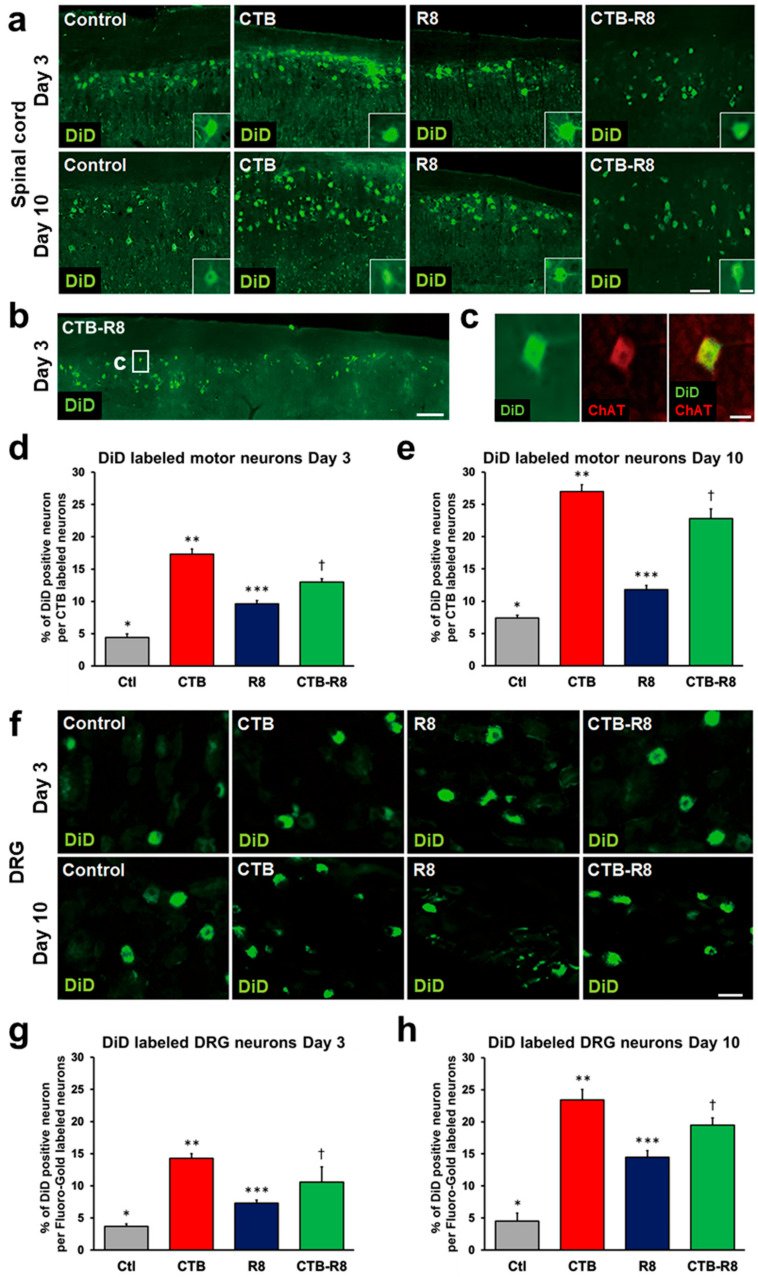
Retrograde transport of liposomes modified with CTB and R8 to spinal cord motor neurons and DRG neurons 3 and 10 days after sciatic nerve injection. (**a**) Horizontal sections of the lumbar spinal cord 3 and 10 days after sciatic nerve injections. Rostral is left. Motor neurons receiving retrogradely transported liposomes were identified by DiD fluorescence. Insets are high-magnification views of DiD-labeled motor neurons. Scale bars, 100 µm, 30 µm (inset). (**b**) Lower-magnification view of the lumbar spinal cord 3 days after sciatic nerve injection of CTB-R8-modified liposomes. Boxed areas indicate locations of (**a**,**c**). Scale bar, 300 µm. (**c**) High-magnification view of (**b**). Immunoreactivity for choline acetyltransferase (ChAT) was detected in a DiD- labeled neuron. Scale bar, 30 µm. (**d**) The efficiency of retrograde transport of liposomes 3 days after sciatic nerve injection. The percentage of DiD- labeled motor neurons of the total number of CTB-labeled motor neurons (*n* = 5/group). CTB-modified liposomes had the highest transport efficiency among the four groups. Data are the mean ± SD. * *p* < 0.05, ** *p* < 0.05, *** *p* < 0.05, † < 0.05 vs. all other groups by one-way ANOVA with the Tukey–Kramer test. (**e**) The efficiency of retrograde transport of liposomes 10 days after sciatic nerve injection. The percentage of DiD-labeled motor neurons of the total number of CTB-labeled motor neurons (*n* = 5/group). Overall transport efficiency was higher than at day 3. Data are the mean ± SD. * *p* < 0.05, ** *p* < 0.05, *** *p* < 0.05, † < 0.05 vs. all other groups by one-way ANOVA with the Tukey–Kramer test. (**f**) L5 DRG sections. DRG neurons receiving retrogradely transported liposomes were identified by DiD fluorescence. Scale bar, 40 µm. (**g**) The efficiency of retrograde transport of liposomes to DRG neurons 3 days after sciatic nerve injection. The percentage of DiD-labeled DRG neurons 3 days after sciatic nerve injection of the total number of Fluoro-Gold-labeled neurons was quantified (*n* = 5/group). CTB-modified liposomes had the highest transport efficiency among the four groups. Data are the mean ± SD. * *p* < 0.05, ** *p* < 0.05, *** *p* < 0.05, † < 0.05 vs. all other groups by one-way ANOVA with the Tukey–Kramer test. (**h**) The efficiency of retrograde transport of liposomes to DRG neurons 10 days after sciatic nerve injection. The percentage of DiD-labeled DRG neurons 10 days after sciatic nerve injection of the total number of Fluoro-Gold-labeled neurons was quantified (*n* = 5/group). Overall transport efficiency was higher than at day 3. Data are the mean ± SD. * *p* < 0.05, ** *p* < 0.05, *** *p* < 0.05, † < 0.05 vs. all other groups by one-way ANOVA with the Tukey–Kramer test.

**Figure 6 ijms-23-06661-f006:**
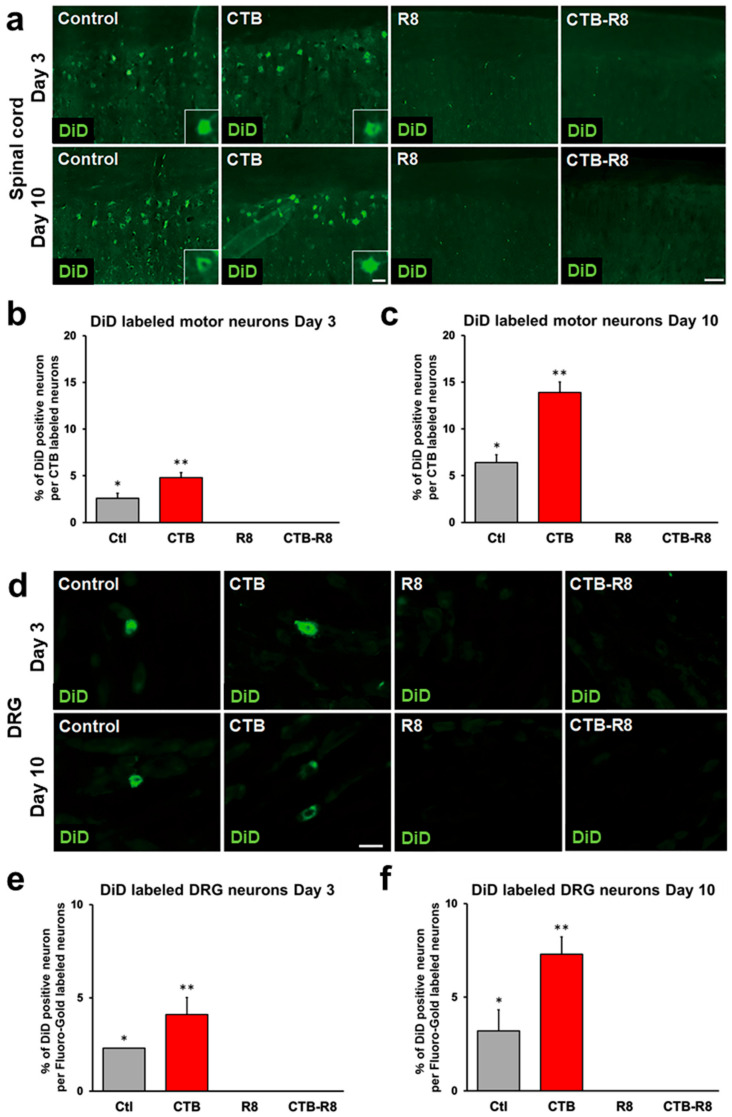
Retrograde transport of liposomes modified with CTB and R8 to spinal cord motor neurons and DRG neurons 3 and 10 days after tibialis anterior injection. (**a**) Horizontal sections of the lumbar spinal cord. Rostral is left. Motor neurons receiving retrogradely transported liposomes were identified by DiD fluorescence. Insets are high-magnification views of DiD-labeled motor neurons. CTB-modified liposomes were detected in motor neurons, whereas R8-modified liposomes were not. Scale bars, 100 µm, 30 µm (inset). (**b**) The efficiency of retrograde transport of liposomes 3 days after tibialis anterior injection. The percentage of DiD-labeled motor neurons of the total number of CTB-labeled motor neurons (*n* = 5/group). CTB-modified liposomes had the highest transport efficiency among the four groups. R8- and CTB-R8-modified liposomes were not identified in motor neurons at all. Data are the mean ± SD. * *p* < 0.05, ** *p* < 0.05 vs. all other groups by one-way ANOVA with the Tukey–Kramer test. (**c**) The efficiency of retrograde transport of liposomes 10 days after tibialis anterior injection. The percentage of DiD-labeled motor neurons of the total number of CTB-labeled motor neurons (*n* = 5/group). Overall transport efficiency was higher than at day 3. Data are the mean ± SD. * *p* < 0.05, ** *p* < 0.05 vs. all other groups by one-way ANOVA with the Tukey–Kramer test. (**d**) L4 DRG sections. DRG neurons receiving retrogradely transported liposomes were identified by DiD fluorescence. Scale bar, 40 µm. (**e**) The percentage of DiD-labeled DRG neurons 3 days after tibialis anterior injection of the total number of Fluoro-Gold-labeled neurons was quantified (*n* = 5/group). CTB-modified liposomes had the highest transport efficiency among the four groups. R8- and CTB-R8-modified liposomes were not identified in motor neurons at all. Data are the mean ± SD. * *p* < 0.05, ** *p* < 0.05 vs. all other groups by one-way ANOVA with the Tukey–Kramer test. (**f**) The percentage of DiD-labeled DRG neurons 10 days after tibialis anterior injection of the total number of Fluoro-Gold-labeled neurons was quantified (*n* = 5/group). Overall transport efficiency was higher than at day 3. Data are the mean ± SD. * *p* < 0.05, ** *p* < 0.05 vs. all other groups by one-way ANOVA with the Tukey–Kramer test.

**Table 1 ijms-23-06661-t001:** The characteristics of liposomes.

Liposome	Composition	Diameter (nm)	PDI	Zeta-Potential (mV)	Lipid Concentration (mM)
DSPC	DSPC/Chol/DSPE-PEG2000 = 70/30/5	136.00 ± 9.27	0.26 ± 0.01	−15.77 ± 0.31	4.73 ± 0.77
POPC	POPC/Chol/DSPE-PEG2000 = 70/30/5	146.67 ± 13.07	0.13 ± 0.01	−16.10 ± 2.62	5.91 ± 0.25
DSPC/POPC	DSPC/POPC/Chol/DSPE-PEG2000 = 30/40/30/5	154.33 ± 14.52	0.15 ± 0.08	−16.57 ± 3.69	5.27 ± 0.17
CTB-DSPC	DSPC/Chol/DSPE-PEG2000 = 70/30/5 CTB modification	173.00 ± 20.61	0.23 ± 0.03	−11.6 ± 2.36	3.68 ± 0.26
R8-DSPC	DSPC/Chol/DSPE-PEG2000 = 70/30/5 R8 modification	116.67 ± 13.07	0.20 ± 0.03	17.9 ± 1.85	4.49 ± 0.23
CTB-R8-DSPC	DSPC/Chol/DSPE-PEG2000 = 70/30/5 CTB + R8 modification	125.00 ± 6.16	0.23 ± 0.03	16.3 ± 1.44	4.2 ± 0.42

Diameter, PDI, and Zeta-potential of liposomes are presented. Data are the mean ± SD (*n* = 3).

## Data Availability

The data generated by the current study are available from the corresponding author upon reasonable request.

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
