# Peer review of "Retrograde Axonal Transport of Liposomes from Peripheral Tissue to Spinal Cord and DRGs by Optimized Phospholipid and CTB Modification"

_ijms, 2022, doi:10.3390/ijms23126661_

Round 1

Reviewer 1 Report

The work entitled “Retrograde axonal transport of liposomes from peripheral tissue to spinal cord and DRGs by optimized phospholipid and CTB modification” by Fukui et all reports on the ability of liposomes to be employed in retrograde transport in axons as a drug delivery system from peripheral tissues to the spinal cord and dorsal root ganglia. Data demonstrated such potentiality. The work is well introduced and the subject is clearly identified as well as the main goal of this research and the authors’ hypothesis. The novelty is also clear. The methodology is well detailed, which allows for other authors to replicate the experiments proposed. Data is scientifically sound and the discussion is very complete, and supported by the literature. There are only two small details that should be addressed prior to publication:

-          The schematic representations made by the authors were assisted with any imaging software? If so, it should be identified.

-          There are some English writing mistakes that should be fixed.

Author Response

Comment 1: The schematic representations made by the authors were assisted with any imaging software? If so, it should be identified.

Response: We didn't use any special imaging software to analyze fluorescence intensity. The presence of the infrared fluorescent signal in neuronal cell body was apparently detected.

Comment 2: There are some English writing mistakes that should be fixed.

Response: We fixed 9 grammatical errors.

Reviewer 2 Report

The current manuscript serves as a significant study for monitoring transport of drug delivery systems, in this case liposomes, to the spinal cord. The manuscript is scientifically sound and of high interest to the readers of IJMS, as it serves as an important step towards optimising drug delivery systems for optimal axonal transport. I have a few minor comments to consider prior to acceptance for publication:

1. The abstract is well written, but lacks details on how retrograde transport of liposomes was monitored - some brief details relating to the experimental methodology would be good here.

2. Have liposomes been used previously for retrograde transport? In the current introduction, it is not clear if this DDS has been used previously or not. I suspect they have been, so if that is the case, it would be good for the authors to include some background about what has been done and what the key conclusions were. If this is the first study ever to monitor retrograde transport with liposomes then that should be made clear.

3. The authors should provide a brief rationale behind the composition of liposomes selected to be investigated in this study. Not all readers of IJMS will be well-experienced with liposomes, so explaining the difference between each lipids may have importance.

4. Since the authors presents the R&D section before methodology, greater detail is required in Table 1 to describe the difference in liposome compositions between each liposome type.

5. The authors inject 2 uL of liposomes. What is the lipid concentration injected? The authors should also consider quantifying the particle concentration (i.e. particles/mL) of each liposome type, using nanoparticle tracking analysis.

6. Does the relative fluorescence intensity change between each liposome formulation?

Reviewer 3 Report

The authors have written a very interesting article on the retrograde transport of liposome nanoparticles. The article is very well written overall and this reviewer just has several comments to help the paper be a little more clear.

Minor

1)section 2.3 is titled r8 modifications but the first 8 lines only discuss CTB

2)Lines276-282 are formated as figure legends and not as normal text

3)the inhibition from CTB and R8 could mae sense. but why does R8 (figure 6) inhibit uptake of the liposomes alone?

7) please add the animal ethics statement and the certifications in which you state that you followed animal procedures .

8) lines 414-418 the authors claim the liposomes were "srurface modified". however, this is not true. the ligans were added into the mixture of the formulation not post modified on the surface. The ligand could be anywhere throughout the matrix or within the liquid core. Please correct this phrasing

Major

The authors did a very good job at comparing two ligands and the mixture between them. Unfortunately R8 was not as promising as hoped and had deleterious effects. I would suggest the authors change the conclusion and discussion to promote CTB while discussing the negatives of R8 instead of still promoting both as promising ligands. You have shown very well that R8 is not a prime candidate and the discussion does not exemplify this. 

Author Response

Comment 1: section 2.3 is titled r8 modifications but the first 8 lines only discuss CTB

Response: R8 was added in the tile of section 2.3.

Comment 2: Lines276-282 are formatted as figure legends and not as normal text

Response: Fixed.

Comment 3: the inhibition from CTB and R8 could make sense. but why does R8 (figure 6) inhibit uptake of the liposomes alone?

Response: We discussed the potential mechanism why R8 modification abolished the retrograde transport ability of DSPC liposomes in Discussion (L417-429).

Comment 4: please add the animal ethics statement and the certifications in which you state that you followed animal procedures.

Response: Because the approved study number and the adherence to the guidelines were already described at the last section as Institutional Review Board Statement after Conclusion, we did not add this information in Method to avoid redundancy.

Comment 5: lines 414-418 the authors claim the liposomes were "surface modified". however, this is not true. the ligands were added into the mixture of the formulation not post modified on the surface. The ligand could be anywhere throughout the matrix or within the liquid core. Please correct this phrasing

Response: As pointed out, R8 was mixed with the formulation. So, we corrected the texts in Methods and graphical abstract accordingly. Regarding CTB, it was attached to the surface of liposomes.

Comment 6: The authors did a very good job at comparing two ligands and the mixture between them. Unfortunately, R8 was not as promising as hoped and had deleterious effects. I would suggest the authors change the conclusion and discussion to promote CTB while discussing the negatives of R8 instead of still promoting both as promising ligands. You have shown very well that R8 is not a prime candidate and the discussion does not exemplify this.

Response: R8 improved the transport efficiency via nerve administration but abolished that via muscle administration, indicating that the efficacy of R8 depends on the administration route. Thus, we believe R8 still has the potential as a modification ligand for nerve administration. In the Abstract, the fact that R8 abolished the transport capability via muscle administration was already stated. In 5. Conclusion, we corrected the texts to make this point clearer to readers.